# Association between Prenatal Exposure to Household Pesticides and Neonatal Weight and Length Growth in the Japan Environment and Children’s Study

**DOI:** 10.3390/ijerph17124608

**Published:** 2020-06-26

**Authors:** Taro Matsuki, Takeshi Ebara, Hazuki Tamada, Yuki Ito, Yasuyuki Yamada, Hirohisa Kano, Takahiro Kurihara, Hirotaka Sato, Sayaka Kato, Shinji Saitoh, Mayumi Sugiura-Ogasawara, Michihiro Kamijima

**Affiliations:** 1Department of Occupational and Environmental Health, Graduate School of Medical Sciences, Nagoya City University, Mizuho-ku, Nagoya, Aichi 4678601, Japan; ebara@med.nagoya-cu.ac.jp (T.E.); h-tamada@med.nagoya-cu.ac.jp (H.T.); yukey@med.nagoya-cu.ac.jp (Y.I.); h-kano@med.nagoya-cu.ac.jp (H.K.); kurihara.81818181@gmail.com (T.K.); h.sato@med.nagoya-cu.ac.jp (H.S.); s-ichiki@med.nagoya-cu.ac.jp (S.K.); kamijima@med.nagoya-cu.ac.jp (M.K.); 2Graduate School of Health and Sports Science, Juntendo University, Inzai, Chiba 2701695, Japan; yayamada@juntendo.ac.jp; 3Department of Pediatrics and Neonatology, Graduate School of Medical Sciences, Nagoya City University, Mizuho-ku, Nagoya, Aichi 4678601, Japan; ss11@med.nagoya-cu.ac.jp; 4Department of Obstetrics and Gynecology, Graduate School of Medical Sciences, Nagoya City University, Mizuho-ku, Nagoya, Aichi 4678601, Japan; og.mym@med.nagoya-cu.ac.jp

**Keywords:** pesticides, pregnancy, children, birth weight, birth length, JECS

## Abstract

The effects of prenatal exposure to household pesticides on fetal and neonatal growth have not been fully clarified. The present study aims to determine the effects of prenatal exposure to pesticides on neonates’ body size and growth during the first month. This study included 93,718 pairs of pregnant women and their children from the Japan Environment and Children’s Study. Participants completed self-reporting questionnaires during their second or third trimesters on their demographic characteristics and frequency of pesticide use during pregnancy. Child weight, length, and sex were obtained from medical record transcripts. Birth weight and length, as well as weight and length changes over the first month, were estimated using an analysis of covariance. Frequency of exposure to almost all pesticides had no effects on birth weight and length. However, we found small but significant associations (i) between the use of fumigation insecticides and decreased birth weight, and (ii) between frequencies of exposure to pyrethroid pesticides, especially mosquito coils/mats, and suppression of neonatal length growth. Prenatal exposure to household pesticides, especially those containing pyrethroids, might adversely influence fetal and postnatal growth trajectories.

## 1. Introduction

Household pesticide products are ubiquitous chemicals. Since the use of organochlorine pesticides is strictly regulated due to their toxicological effects and long-term persistence in the environment, alternative pesticides, such as pyrethroids, and organophosphorus insecticides have become prevalent in agriculture [1,2]. Pyrethroids are also the most widely used household pesticides because of their relatively low toxicity to mammals [3]. Consequently, people within the general population are commonly exposed to these chemicals, and the number of studies addressing the potential health risks of these exposures has been increasing [4,5,6,7].

A public health concern regarding household pesticides is whether these exposures exert any negative effects on children’s health and development. For example, animal studies have reported that pyrethroids may have thyroid-disrupting potencies [8,9]. Thyroid-disrupting chemicals may interfere with the structure or function of the thyroid gland, the regulatory enzymes associated with thyroid hormone homeostasis, or change circulating or tissue concentrations of thyroid hormones [10]. In human studies, the sum of *cis*-dimethyl-cyclopropane carboxylic acid (*cis*-DCAA), *trans*-DCAA, and 3-PBA was associated with decreased birth weight [6]. Maternal exposure to synthetic pyrethroids in the first or second trimester was associated with a small but statistically significant decrease in birth weight in an epidemiologic study [11]. These findings may reflect the thyroid-disrupting properties of pyrethroids. In contrast, some studies have not found associations between maternal exposure to pyrethroids and birth size (e.g., birth weight) in humans [12,13]. Thus, the potential inhibitory effects of pyrethroids on fetal growth in humans are still unclear.

The developing nervous systems of neonates may also be vulnerable to neurotoxic chemicals [14]. Studies examining associations between pesticide exposure and the growth trajectories of neonates need to be expanded upon to include more comprehensive endpoints. For example, maternal serum p,p’-DDT concentrations have been positively associated with body weight in girls aged 1−2 years, and maternal urinary pyrethroid metabolite concentrations; notably, cis-3-(2,2-dibromovinyl)-2,2-dimethyl-cyclopropane carboxylic acid (cis-DBCA) and trans-DCCA, have been negatively associated with body weight in boys aged 1−2 years [15]. Furthermore, an epidemiologic study found that a 10-fold increase in cis-DCCA concentration was associated with a 0.21 kg/m^2^ lower body mass index at age 3.5 years [16]. However, previous studies have not fully examined the relationship between maternal household pesticide exposure during pregnancy and the growth of neonates, whose neurodevelopment is particularly immature.

Hence, our study aimed to investigate the effects of prenatal exposure to various types of household pesticides on neonatal growth trajectories with regard to body weight and length occurring from birth to the first month of life using the resources of a large, longitudinal Japanese birth cohort study.

## 2. Materials and Methods

### 2.1. Study Design

Pregnant women were recruited for participation in the Japan Environment and Children’s Study (JECS) between 2011 and 2014. JECS is a nationwide birth cohort study investigating environmental factors potentially affecting children’s health and development [17,18,19]. The JECS was approved by the Institutional Review Board of the Japan National Institute for Environmental Studies (no. 100910001), as well as by the ethics committees of all participating institutions. Written informed consent was obtained from all participating women and their partners in accordance with the Declaration of Helsinki. Data sharing is not permitted by the JECS due to a government policy restricting the deposition of data containing personal information, as described in more detail within the references.

### 2.2. Participants

This study was based on the jecs-ag-20160424 dataset, which includes 104,102 records, and was released restrictively to concerned parties in June 2016. Eligibility criteria for expectant mothers were as follows: (1) living in the study area at the time of recruitment (i.e., in any of the 15 selected study areas located throughout Japan); (2) an expected delivery date after August 1, 2011; and (3) comprehension of the Japanese language and the ability to complete the self-administered questionnaire [17,18]. Detailed information regarding the JECS project have been published previously [19].

The following potential participants were excluded from this study: 29 women who dropped out of the JECS, 1994 women with multiple pregnancies, 1530 women with miscarriages and stillbirths, 2290 women with missing birth status data, and 4541 women with missing data from the questionnaire about the use of pesticides. Therefore, 93,718 pairs of mothers and their neonates with singleton live births and complete data regarding question items about pesticide use were included in this study (Figure 1).

### 2.3. Variables

#### 2.3.1. Exposures

Data on the daily use of household pesticides during pregnancy were collected from questionnaires administered between the 2nd and 3rd trimesters (from week 22 to the end of the pregnancy). The classes of pesticides queried in this study are the same as those described in the JECS profile paper [20]. Specifically, our study evaluated the following pesticide exposures: spraying insecticide indoors (never/less than once a month/1−3 times a month/once a week/a few times a week/everyday), mosquito coil or electric mosquito repellant mats used continuously for more than a few hours (never/less than once a month/1−3 times a month/once a week/a few times a week/everyday), use of liquid insecticides for maggots and mosquito larvae (never/less than once a month/1−3 times a month/once a week/a few times a week/everyday), herbicide or gardening pesticide used for gardens, balcony gardens, or farms (never/less than once a month/1−3 times a month/once a week/a few times a week/everyday), insect repellents sprayed on clothes or applied as skin lotion (never/less than once a month/1−3 times a month/once a week/a few times a week/everyday), moth repellent for clothes in storage (never/sometimes/continuously), and smoke-based insecticide administered indoors (no/yes). Answer choices of “less than once a month” and “1−3 times a month” are referred to as “occasionally” throughout the manuscript, and answer choices of “once a week” and “a few times a week” are each referred to as “sometimes”.

The definitions of pesticides evaluated in this study are as follows: (a) indoor insecticide sprays are insecticidal agents sprayed directly onto insects, (b) mosquito coils/mats control mosquitoes by liquid insecticides that evaporate and volatilize due to high temperature/heat, (c) liquid insecticides are insecticidal agents, including those diluted with water, that control fly and mosquito larvae, (d) herbicides/gardening pesticides either kill weed leaves and stems with a weeding drug that affects the roots, or control pests by spraying pesticidal agents diluted with water, (e) insect repellents repel pests by spraying the exposed areas of the clothes or skin, (f) moth repellents for clothes repel pests attached to skin or clothes that are in storage, and (g) fumigation insecticides exterminate pests by pesticides within smoke rather than sprays, lotions, etc.

#### 2.3.2. Outcomes

The body size data of neonates were obtained from medical record transcripts. The variables evaluated as outcomes in this study are as follows: birth weight (g), birth length (cm), weight change during the first month after birth (weight change; g) and length change during the first month after birth (length change; cm).

#### 2.3.3. Covariates

Data regarding covariates were used in analysis of covariance (ANCOVA). Among the variables included in the JECS questionnaire, we used the covariates reported in previous studies that were associated with the exposures or outcomes used in this study [21,22]. The variables used as covariates are as follows: sex of neonate (male/female), age of mother (<20; 20–29; 30–39; ≥40), body mass index before pregnancy, i.e., kg/m^2^ (<18.5; 18.5–24.9; ≥25.0), weight change during pregnancy (kg; continuous), placental weight (g; continuous), parity (0; ≥1), smoking history (nonsmokers/exsmokers who quit before pregnancy/exsmokers who quit during early pregnancy/current smokers), alcohol consumption (nondrinkers/exdrinkers who quit before pregnancy/exdrinkers who quit during early pregnancy/current drinkers), gestational age (day; continuous), annual household income in millions of Japanese yen (<2; 2–<4; 4–<6; 6–<8; 8–<10; ≥10), gestational diabetes (no/yes), hypertensive disorders of pregnancy (mild or severe; no/yes), mode of delivery (spontaneous delivery/induced delivery/vacuum extraction or forceps delivery/cesarean delivery), and the use of other pesticides that were beyond the focus of this analysis. Since the type and frequency of pesticides used may vary depending on the area of residence and the season, participants’ residential area was classified into one of the mean annual temperature zones (<9.0 °C; 11.9–13.0 °C; 14.1–14.9 °C; 15.7–15.9 °C; 16.7–16.9 °C; 17.0–17.4 °C; ≥23.0 °C) based on statistics collected between 1981−2010 by the Japan Meteorological Agency, and the date of entry into the questionnaire about pesticide use (January–February; March–April; May–June; July–August; September–October; November–December).

In ANCOVA analyses evaluating weight change and length change as dependent variables, we included birth weight in the weight change analyses and birth length as covariates in the length change analyses. We also included any conditions for which the children in the study were currently receiving medical treatment (no/yes), feeding methods (breastfeeding only/mixed feeding/infant formula only) and the results of congenital metabolic disorder tests (normal/reexamination needed/detailed examination needed/diagnosis confirmed) as covariates.

### 2.4. Statistical Methods

Multiple correspondence analysis (MCA) and hierarchical cluster analysis (HCA; Ward’s method) were used to investigate connections between the use of seven types of pesticides, residential area and date of completing the pesticide questionnaire.

We used ANCOVA to evaluate differences in birth weight and birth length, as well as weight change and length change during the first month after birth, according to the level of pesticide exposure during pregnancy. Pairwise comparisons were made using Bonferroni post hoc analysis.

To examine the effects of exposures to multiple pesticides, we also examined the relationship between use patterns for the top ten most frequently used pesticides among the study sample and child-body sizes. Specifically, we determined 128 pesticide use patterns based on answers of “at least one used” and “never used” for the seven pesticides and analyzed the ten most prevalent use patterns. All analyses were conducted using SPSS version 24.0 (IBM Corp., Armonk, NY, USA).

## 3. Results

### 3.1. Maternal and Neonate Characteristics

Table 1 shows maternal and neonate characteristics. Overall, the mean (±SD; standard deviation) birth weights among children born to mothers who did not use pesticides and who did use pesticides during pregnancy were 3020.8 (±413.0) g and 3027.8 (±413.2) g, respectively. The corresponding mean (±SD) birth lengths were 48.9 (±2.2) cm and 49.0 (±2.2) cm, respectively. The percentages of women who used indoor insecticide sprays, mosquito coils/mats, liquid insecticides, herbicides/gardening pesticides, insect repellents on clothes/skin, moth repellents for clothes and fumigation insecticides while pregnant were 32.3%, 31.9%, 0.7%, 8.7%, 24.7%, 59.0% and 6.7%, respectively (*n* = 93,718).

### 3.2. Pesticide Use Profiles during Pregnancy

Data regarding pesticide use during pregnancy according to MCA are shown on the X–Y plane in Figure 2; pesticide use profiles were categorized into four clusters by HCA. Cluster A consisted of pesticide use during autumn and winter (September to December), a higher than normal temperature (≥23 °C) of the residential area, and participants in the “occasionally” groups for three pesticide categories (indoor insecticide sprays, mosquito coils/mats and insect repellents on clothes/skin). Cluster B consisted of pesticide use during the summer season (July to August) and the “occasionally” to “everyday” groups with regard to the use of pesticides (indoor insecticide sprays, mosquito coils/mats, liquid insecticides, herbicides/gardening pesticides and insect repellents on clothes/skin). Cluster C consisted of the “sometimes” to “everyday” groups with regard to four pesticides (indoor insecticide sprays, liquid insecticides, herbicides/gardening pesticides and insect repellents on clothes/skin). Finally, cluster D consisted of all the remaining items except the “everyday” group with regard to liquid insecticide use.

### 3.3. Associations with Birth Weight and Birth Length

Differences in birth weight and length according to the frequency of pesticide use during pregnancy were examined with ANCOVA followed by a Bonferroni post hoc test (Table 2). Due to missing covariates within a complete case analysis approach, the sample size for birth weight analyses was *n* = 71,116, and that for birth length analyses was *n* = 71,019. For analyses examining the use of mosquito coils/mats, average birth length was smaller in the “sometimes” group than in the “never” group. For analyses examining the use of moth repellents for clothes, average birth length was larger in the “sometimes” and “continuously” groups than in the “never” group. Finally, for analyses examining the use of fumigation insecticides, birth weight was smaller in the “yes” group than in the “no” group. These effect sizes (*η_p_^2^*) ranged from <0.00001 to 0.00013 for birth weight and <0.00001 to 0.00021 for birth length. The maximum statistically significant difference between the observed mean values for the nonexposed groups (never/no) and the exposed groups (occasionally/sometimes/everyday/continuously/yes) for each pesticide was 11.55 g difference in birth weight with regard to use of fumigation insecticides, and 0.06 cm difference in birth length with regard to the use of mosquito coils/mats.

### 3.4. Associations with Weight Change and Length Change

We examined differences in weight and length changes according to the frequency of pesticide use during pregnancy with ANCOVA followed by a Bonferroni post hoc test (Table 3). Due to missing covariates within a complete case analysis approach, sample sizes for examining weight change and length change were *n* = 68,586 and *n* = 67,932, respectively. For analyses examining the use of mosquito coils/mats, length change was smaller in the “occasionally”, “sometimes” and “everyday” groups than in the “never” group. For analyses examining the use of insect repellents on clothes/skin, weight change was smaller in the “occasionally” group than in the “never” group; length change was also smaller in the “occasionally” and “sometimes” groups than in the “never” group. These effect sizes (*η_p_^2^*) ranged from 0.00002 to 0.00024 with regard to weight change, and from <0.00001 to 0.00059 with regard to length change. The maximum statistically significant difference between the observed mean values of the nonexposed groups (never/no) and the exposed groups (occasionally/sometimes/everyday/continuously/yes) for each pesticide was 14.36 g difference in weight change with regard to the use of insect repellents on clothes/skin, and 0.11 cm difference in length change with regard to the use of mosquito coils/mats.

### 3.5. Associations between Pesticide Use Patterns and Neonate Body Sizes

When analyzing all 128 pesticide usage patterns, the use of three classes of pesticides (liquid insecticides, herbicides/gardening pesticides and fumigation insecticides) was not confirmed within the top ten usage patterns, so these pesticides were excluded from the analysis (Table 4). Regarding birth length, the most prevalent use pattern (only using moth repellents for clothes) was associated with a greater birth length than the second most prevalent usage pattern (not using any pesticides). Regarding length change, the seventh (using indoor insecticide sprays, mosquito coils/mats and moth repellents for clothes), eighth (using only mosquito coils/mats), and ninth (using mosquito coils/mats, insect repellents on clothes/skin and moth repellents for clothes) most prevalent usage patterns were associated with smaller length changes relative to the first, second and sixth (only using indoor insecticide sprays) most prevalent patterns. In addition, length change for the fifth most prevalent pattern (using indoor insecticide sprays, mosquito coils/mats, insect repellents on clothes/skin and moth repellents for clothes) was smaller than for the first, second, third, fourth and sixth most prevalent use patterns.

### 3.6. Additional Analyses

In addition, we calculated the ratio of weight change divided by birth weight and the ratio of birth length change divided by birth length; significant associations with these measures were observed only for the use of mosquito coils/mats and the use of insect repellents on clothes/skin (Appendix A). As a result, the effects of mosquito coil/mat use and insect repellent use on clothes/skin were statistically significant with regard to the length change ratio. Furthermore, to examine whether placental weight was affected by prenatal pesticide exposure, we performed a covariance analysis with placental weight as the outcome and sex of neonate, age of mother, body mass index before pregnancy, weight change during pregnancy, parity, smoking history, alcohol consumption, gestational age, annual household income, gestational diabetes, hypertensive disorders of pregnancy, mode of delivery, residential area and date of entry into the questionnaire about pesticide use as covariates, but no significant intergroup differences were observed for any pesticide (Appendix A).

### 3.7. Sensitivity Analysis

We examined the differences in characteristics between those who were excluded from this study (*n* = 10,384) and those who were included in this study (*n* = 93,718). T-tests showed significant differences in age (target group vs. excluded group: 30.75 ± 5.03 vs. 30.54 ± 5.46, *t* = 3.38, *p* < 0.01, *q* < 0.01, *d* = 0.04) and body mass index before pregnancy (target group vs. excluded group: 21.23 ± 3.30 vs. 21.31 ± 3.45, *t* = 2.11, *p* = 0.04, *q* = 0.11, *d* = 0.02).

## 4. Discussion

### 4.1. Profile of Pesticide Use during Pregnancy

The pesticide use pattern identified as cluster A suggests that, in areas where normal temperatures are relatively high, pesticides commonly used in the summer (indoor insecticide sprays, mosquito coils/mats and insect repellents for clothes/skin) might be occasionally used even after summer. With respect to these three pesticide types, reductions in length change were observed in each “occasionally” group. The use of these pesticides might therefore have a negative impact, though the magnitude of the effects on child length at the individual level was negligible; rather, these changes in effect may have significance on a population level. The pesticide use pattern identified as cluster B suggests that the use of mosquito coils/mats might increase in the summer. Based on the results of ANCOVA in this study, mosquito coils/mats containing pyrethroids as the main component had a relatively stronger effect of suppressing child length change compared to other pesticides; therefore, it is necessary to pay more attention to the use of mosquito coils/mats among pregnant women based on concerns regarding growth trajectories for newborns. The pesticide use pattern identified as cluster C in our study suggests that indoor insecticide sprays and insect repellents used on clothes/skin are used more frequently in conjunction with herbicide/gardening pesticides. Such exposure patterns can occur when house gardening is performed outside. Future studies are needed to examine in detail the relationship between pesticide exposure status according to occupation and child body sizes. Finally, the pesticide use pattern identified as cluster D, shown near the origin of the X–Y plane, was excluded from the discussion because it comprises an essentially unrelated set of items. Hence, it may be better to consider the effects of residential area and season when considering pesticide exposure.

### 4.2. Differences in the Child Body Sizes

Indoor insecticide spray use was not associated with birth body sizes and child body growth one month after birth in our study. Many indoor insecticide sprays in Japan contain pyrethroids; however, we found that the use of indoor insecticide spray had no effect on child body development. Nevertheless, we acknowledge the possibility that some mothers in our study mistook freezing sprays, which stop the movement of insects such as cockroaches, for spray insecticides, leading to exposure misclassification. Most freezing sprays do not contain pyrethroids, so it is difficult to determine the effects of spray insecticides in this study.

Mosquito coils/mats, for which pyrethroid is the main component, were not associated with infant body weight, but were associated with the suppression of birth length and length growth after birth. A dose–response relationship was observed, such that postnatal length development was suppressed with an increasing frequency of mosquito coil/mat use. Unlike indoor insecticide sprays, mosquito coils/mats are filled with insecticidal components such as pyrethroids that volatilize efficiently; therefore, pregnant women may be more likely to be exposed to pyrethroid through this route than via indoor insecticide sprays.

Approximately 1% of our study population comprised liquid insecticide users. The usage rate of herbicide/gardening pesticides was also low, with a prevalence of approximately 9%. Therefore, our study lacked sufficient statistical power to examine associations between these pesticides and weight, length and growth trajectory parameters. Although relatively few people used these classes of pesticides, the birth weight of infants born among the “everyday” group was less than 3000 g for mothers who used liquid insecticides or herbicides/gardening pesticides during pregnancy. Therefore, future studies should prioritize evaluating the potential effects of these pesticides on child body sizes.

Our results suggest that although insect repellents on clothes/skin were not associated with birth weight and length, this pesticide may suppress child body growth one month after birth. Although DEET has been considered relatively safe for humans [23], a study observed fetal birth weight loss in the litters of pregnant rats exposed to high doses of DEET [24]. The impact of the effects of DEET on human thyroid hormone remains unclear, and the association between DEET exposure and human child body sizes and growth trajectories requires careful interpretation.

The positive association we observed between moth repellents for clothes and birth length growth trajectory may be related to the effect of paradichlorobenzene (*p*-DCB), the main component of clothing insect repellents. It has been suggested that urinary 2,5-dichlorophenol (2,5-DCP), a reliable biomarker for measuring *p*-DCB exposure, may contribute to obesity in human children and adults [25,26]. *p*-DCB has thyroid-disrupting properties [27], and the specific resulting disruption is likely to be associated with obesity. On the other hand, it is difficult to conclude that this result was an effect of only *p*-DCB, because there are some kinds of moth repellents for clothes containing pyrethroid. Our findings that *p*-DCB, which may be associated with obesity, is associated with birth length but not birth weight needs to be examined further in future research.

Fumigation insecticide use was associated with a decrease in birth weight in our study. This may have resulted from exposure to pyrethroids, though fumigation insecticides also contain oxadiazole as a main component in addition to pyrethroid pesticides. Regarding effects on birth weight, oxadiazole may inhibit diacylglycerol *O*-acyltransferase 1, which is involved in the production of neutral fat [28]. Maternal smoking during pregnancy was also found to reduce birth weight by about 4% in a study among the JECS cohort [21]. The effect caused by fumigation insecticide in our study was about 1/10 of the suppressive effect of maternal smoking. The ingredients and mechanisms potentially mediating associations between birth weight and fumigation insecticide use in our study need to be addressed further in future research.

Examination of combined exposures based on the 10 most prevalent pesticide usage patterns showed that use of mosquito coils/mats was associated with a decreased length change growth trajectory. For participants using only mosquito coils/mats and moth repellents for clothes, the effects of mosquito coils/mats may have been offset by the effects of paradichlorobenzene found in moth repellents for clothes. Of note is that child length change was minimal when four pesticides (indoor insecticide sprays, mosquito coils/mats, insect repellents for clothes/skin, and moth repellents for clothes) were used in combination. The multiple combined effects of the active ingredients in these pesticides, such as pyrethroids, DEET, and paradichlorobenzene, on the body sizes of children may be complex, and the effects of multiple human exposures need to be studied in detail.

Only mosquito coils/mats and insect repellents used on clothes/skin showed a statistically significant association with the length change ratio in our study. We also did not observe associations between pesticide exposure and placental weight, which indicates that placental development may not be a mechanism mediating the observed associations between pesticide exposure and infant weight, length and growth trajectories.

An overview of the above results suggests that exposure to pesticides composed mainly of pyrethroids may contribute to the suppression of the body size of neonates. This supports the suggestion from previous studies that fetal pyrethroid exposure may continue to alter children’s thyroid hormone homeostasis even after birth [29]. Although the slight differences observed in the results of this study were almost negligible in terms of biological individual differences, it should be noted that statistical differences were observed that might have impacts at a population level. In the future, researchers may identify more sensitive endpoints than neonates’ length and weight for evaluating impacts on growth and development.

Previous studies indicate that exposure of mothers to pesticides may affect not only the development of the body size of neonates, but also the child’s neurodevelopment relating to cognition and behavior. For example, it has been reported that exposure to pesticides such as pyrethroids during pregnancy increases the risk of neurodevelopmental disorders (e.g., autism spectrum disorder and attention/hyperactivity disorder) [30]. Studies that found associations between low birth weight, poor fetal growth and the risk of neurodevelopmental disorders [31] suggested that maternal exposures to pesticides, child body growth and child neurodevelopment disorders are closely and complexly related. Future studies will be needed to elucidate the impact of pesticide exposure on children’s comprehensive development.

### 4.3. Strengths and Limitations

The subjects of this study were mothers from among the general population, represented by a Japanese birth cohort enrolling nearly 100,000 participants. They are likely to be representative of the general population based on sensitivity analyses comparing included and excluded participants. Thus, we believe the results in this study are robust and have high generalizability. In our opinion, this is the primary strength of our study.

While the self-reported questionnaires used in the JECS can provide some insight regarding dose–response relationships, we could not identify the actual chemicals contained in each pesticide based on this study design, and hence, can only speculate about the mechanisms potentially mediating the effect of each pesticide on child body size. The hypothesized effects of pesticides, such as a possible inhibitory effect of pyrethroid or a possible promoting effect of *p*-DCB on child length development, are uncertain. In our opinion, this is the main limitation of our study. Besides, the JECS administered not only a questionnaire survey, but also estimated pesticide exposure using urinary metabolite assays. Urinary metabolite data will be used to examine further associations between the combined exposure patterns observed in this study and child body sizes and growth trajectories in future research efforts.

## 5. Conclusions

In this study, we found that the frequencies of exposure to almost all the pesticides we evaluated had no effects on birth weight and length. However, we found significant associations between the use of fumigation insecticides and decreased birth weight, and between frequencies of exposure to some pyrethroid-based pesticides and suppressed length growth of neonates, although the effect sizes were small. These results suggested that prenatal exposure to household pesticides, especially containing pyrethroids, such as mosquito coils/mats, might adversely influence fetal and postnatal growth trajectories. The key issue to evaluate further is balancing the benefits and possible risks of the use of household pesticides during pregnancy. Future studies will need to examine how the extent and timing of pesticide use and the specific pesticides used may affect the comprehensive trajectory of child development.

## Figures and Tables

**Figure 1 ijerph-17-04608-f001:**
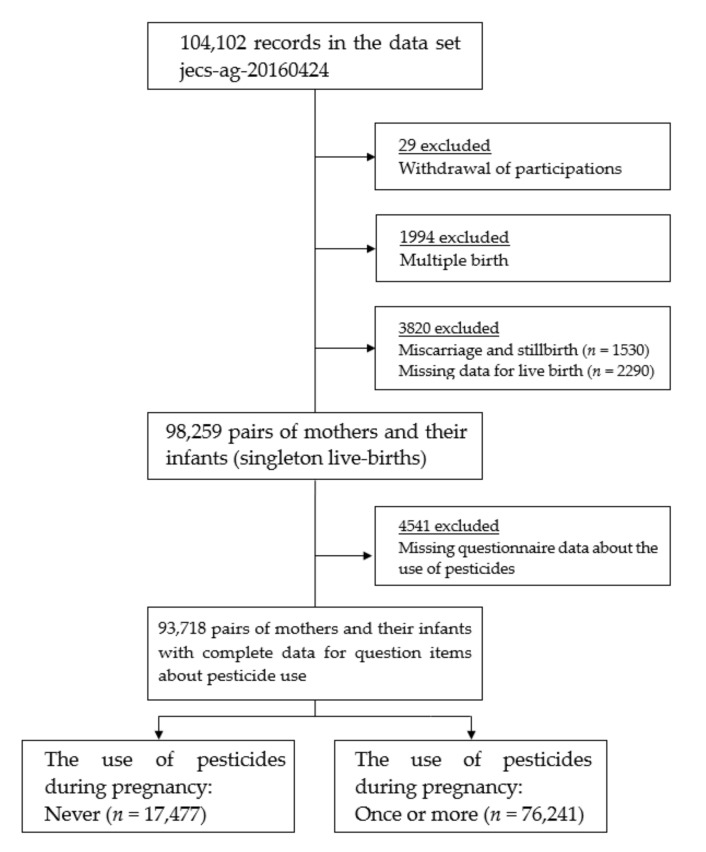
Flow chart of study participants.

**Figure 2 ijerph-17-04608-f002:**
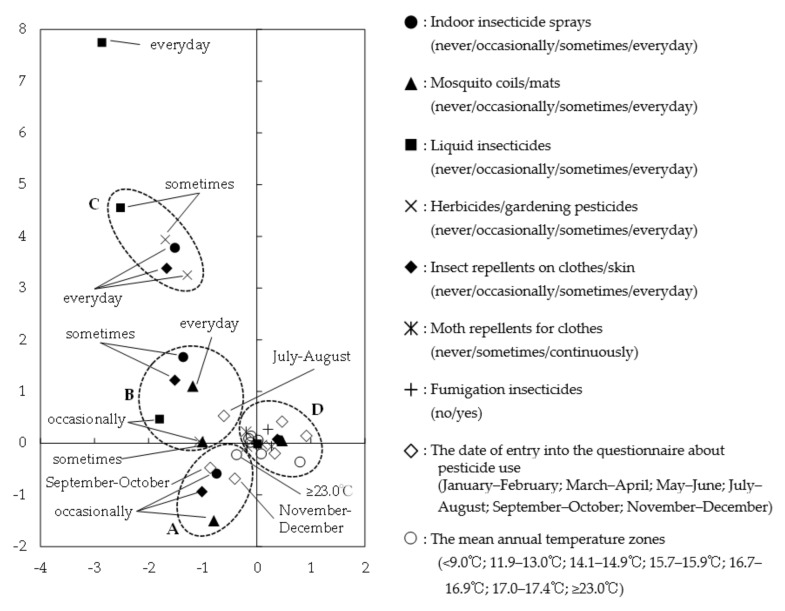
The results of MCA and the hierarchical cluster analysis (Ward method).

**Table 1 ijerph-17-04608-t001:** Maternal and child characteristics.

Variables	Total				The Use of Pesticides				
					Never			Once or More	
	*n* = 93,718				*n* = 17,477			*n* = 76,241	
Children									
Sex	*n*		(%)		*n*	(%)		*n*	(%)
male	48,007		(51.2)		8915	(51.0)		39,092	(51.3)
female	45,707		(48.8)		8560	(49.0)		37,147	(48.7)
missing	4				2			2	
Feeding methods	*n*		(%)		*n*	(%)		*n*	(%)
breastfeeding only	39,201		(41.8)		7490	(42.9)		31,711	(41.6)
mixed feeding	51,614		(55.1)		9340	(53.4)		42,274	(55.4)
infant formula only	1325		(1.4)		295	(1.7)		1030	(1.4)
missing	1578				352			1226	
Body sizes	mean		±SD		mean	±SD		mean	±SD
birth weight (g)	3026.5				3021			3027.76	±413.2
missing	61				11			50	
birth length (cm)	49.0		±2.2		48.9	±2.2		49.0	±2.2
missing	396				67			329	
weight change for a month after birth (g)	1098.9				1107	±374.8		1097.0	±375.8
missing	1363				218			1145	
length change for a month after birth (cm)	4.3		±1.8		4.4	±1.8		4.2	±1.8
missing	2441				404			2037	
Results of congenital metabolic disorder tests	*n*		(%)		*n*	(%)		*n*	(%)
normal	89,436		(95.4)		16,631	(95.2)		72,805	(95.5)
reexamination needed	875		(0.9)		177	(1.0)		698	(0.9)
detailed examination needed	106		(0.1)		20	(0.1)		86	(0.1)
diagnosis confirmed	200		(0.2)		17	(0.1)		183	(0.2)
missing	3101				632			2469	
Disease currently receiving medical treatment	*n*	(%)		*n*		(%)	*n*		(%)
no	80,611	(86.0)		15,063		(86.2)	65,548		(86.0)
yes	9956	(10.6)		1711		(9.8)	8245		(10.8)
missing	3151	(3.4)		703		(4.0)	2448		(3.2)
Mothers									
Age (years)	*n*	(%)		*n*		(%)	*n*		(%)
<20	972	(1.0)		318		(1.8)	654		(0.9)
20−29	34,757	(37.1)		7466		(42.7)	27,291		(35.8)
30−39	49,167	(52.5)		8168		(46.7)	40,999		(53.8)
≥40	3126	(3.3)		509		(2.9)	2617		(3.4)
missing	5696			1016			4680		
Body mass index before pregnancy (kg/m^2^)	*n*	(%)		*n*		(%)	*n*		(%)
<18.5	14,609	(15.6)		2917		(16.7)	11,692		(15.3)
18.5−24.9	68,807	(73.4)		12,536		(71.7)	56,271		(73.8)
≥25.0	10,242	(10.9)		2007		(11.5)	8235		(10.8)
missing	60			17			43		
Weight change during pregnancy (kg)	mean	±SD		mean		±SD	mean		±SD
	10.2	±4.0		10.4		±4.1	10.2		±3.9
missing	7698			1594			6104		
Placental weight (g)	mean	±SD		mean		±SD	mean		±SD
	559.41			559.2			559.47		
missing	3436			720			2716		
Parity	*n*	(%)		*n*		(%)	*n*		(%)
0	36,930	(39.4)		7709		(44.1)	29,221		(38.3)
≥1	54,584	(58.2)		9251		(52.9)	45,333		(59.5)
missing	2204			517			1687		
Smoking history	*n*	(%)		*n*		(%)	*n*		(%)
non-smokers	53,784	(57.4)		9801		(56.1)	43,983		(57.7)
ex-smokers who quit before pregnancy	22,191	(23.7)		3749		(21.5)	18,442		(24.2)
ex-smokers who quit during early pregnancy	12,712	(13.6)		2764		(15.8)	9948		(13.0)
current smokers	4284	(4.6)		1007		(5.8)	3277		(4.3)
missing	747			156			591		
Alcohol consumption	*n*	(%)		*n*		(%)	*n*		(%)
non-drinkers	31,169	(33.3)		5916		(33.9)	25,253		(33.1)
ex-drinkers who quit before pregnancy	15,945	(17.0)		2875		(16.5)	13,070		(17.1)
ex-drinkers who quit during early pregnancy	43,239	(46.1)		8112		(46.4)	35,127		(46.1)
current drinkers	2620	(2.8)		431		(2.5)	2189		(2.9)
missing	745			143			602		
Gestational age (day)	mean	±SD		mean		±SD	mean		±SD
	274.8	±10.7		274.8		±10.8	274.8		±10.6
missing	34			11			23		
Annual household income (million Japanese yen)	*n*	(%)		*n*		(%)	*n*		(%)
<2	4954	(5.3)		1156		(6.6)	3798		(5.0)
2–<4	30,267	(32.3)		5968		(34.1)	24,299		(31.9)
4–<6	28,801	(30.7)		5136		(29.4)	23,665		(31.0)
6–<8	13,912	(14.8)		2413		(13.8)	11,499		(15.1)
8–<10	5687	(6.1)		965		(5.5)	4722		(6.2)
≥10	3718	(4.0)		581		(3.3)	3137		(4.1)
missing	6379			1258			5121		
Gestational diabetes	*n*	(%)		*n*		(%)	*n*		(%)
no	91,186	(97.3)		17,048		(97.5)	74,138		(97.2)
yes	2532	(2.7)		429		(2.5)	2103		(2.8)
missing	0			0			0		
Hypertensive disorders of pregnancy	*n*	(%)		*n*		(%)	*n*		(%)
no	90,834	(96.9)		16,947		(97.0)	73,887		(96.9)
yes	2884	(3.1)		530		(3.0)	2354		(3.1)
missing	0			0			0		
Mode of delivery	*n*	(%)		*n*		(%)	*n*		(%)
spontaneous delivery	53,897	(57.5)		9838		(56.3)	44,059		(57.8)
induced delivery	16,639	(17.8)		3296		(18.9)	13,343		(17.5)
forceps delivery/vacuum extraction	5408	(5.8)		1040		(6.0)	4368		(5.7)
cesarean delivery	17,547	(18.7)		3261		(18.7)	14,286		(18.7)
missing	227			42			185		
The date of completing the questionnaire	*n*	(%)		*n*		(%)	*n*		(%)
January–February	14,662	(15.6)		3462		(19.8)	11,200		(14.7)
March–April	15,835	(16.9)		4961		(28.4)	10,874		(14.3)
May–June	18,491	(19.7)		4116		(23.6)	14,375		(18.9)
July–August	16,843	(18.0)		1900		(10.9)	14,943		(19.6)
September–October	14,287	(15.2)		1290		(7.4)	12,997		(17.0)
November–December	13,600	(14.5)		1748		(10.0)	11,852		(15.5)
missing	0			0			0		
Normal temperature by residential area	*n*	(%)		*n*		(%)	*n*		(%)
<9.0℃ (Hokkaido)	7477	(8.0)		2978		(17.0)	4499		(5.9)
11.9−13.0℃ (Miyagi, Fukushima and Nagano)	22,905	(24.4)		4006		(22.9)	18,899		(24.8)
14.1−14.9℃ (Yamanashi, Toyama and Tottori)	12,154	(13.0)		2074		(11.9)	10,080		(13.2)
15.7−15.9℃ (Chiba, Kanagawa, Aichi and Kyoto)	19,935	(21.3)		3011		(17.2)	16,924		(22.2)
16.7−16.9℃ (Osaka, Hyogo and Kumamoto)	14,978	(16.0)		2453		(14.0)	12,525		(16.4)
17.0−17.4℃ (Kochi, Fukuoka and Miyazaki)	15,450	(16.5)		2843		(16.3)	12,607		(16.5)
≥23.0℃ (Okinawa)	819	(0.9)		112		(0.6)	707		(0.9)
missing	0			0			0		

SD: standard deviation.

**Table 2 ijerph-17-04608-t002:** Differences in neonate body sizes (birth weight and birth length) according to the frequency of pesticide use during pregnancy.

Pesticides	Frequency of Use		Birth Weight (g)							Birth Length (cm)						
			*n*	Mean ^1^	SE ^2^	*p*	*η* _p_ ^2^	*q*	Post Hoc ^3^	*n*	Mean ^1^	SE ^2^	*p*	*η* _p_ ^2^	*q*	Post Hoc ^3^
(a) Indoor insecticide sprays	1	never	47,889	3031.36	1.19	0.45	0.00004	0.59		47,823	48.98	0.01	0.02	0.00014	0.08	
	2	occasionally	18,917	3032.17	1.91					18,895	48.95	0.01				
	3	sometimes	3902	3028.35	4.19					3894	48.93	0.03				
	4	everyday	408	3049.05	12.73					407	49.11	0.08				
(b) Mosquito coils/mats	1	never	48362	3032.04	1.20	0.17	0.00007	0.34		48,288	48.98	0.01	<0.01	0.00019	0.02	
	2	occasionally	6773	3032.87	3.14					6768	48.93	0.02				
	3	sometimes	9538	3026.20	2.69					9527	48.92	0.02				3 < 1
	4	everyday	6443	3033.98	3.28					6436	48.97	0.02				
(c) Liquid insecticides	1	never	70,635	3031.56	0.96	0.32	0.00005	0.52		70,538	48.97	0.01	0.97	<0.00001	0.97	
	2	occasionally	392	3030.12	12.97					392	48.97	0.08				
	3	sometimes	71	3018.00	30.42					71	49.06	0.20				
	4	everyday	18	2921.75	60.34					18	48.99	0.39				
(d) Herbicides/gardening pesticides	1	never	64,863	3031.59	1.01	0.30	0.00005	0.52		64,776	48.97	0.01	0.42	0.00004	0.56	
	2	occasionally	5898	3030.77	3.38					5888	48.97	0.02				
	3	sometimes	288	3042.52	15.11					288	49.13	0.10				
	4	everyday	67	2976.94	31.28					67	49.03	0.20				
(e) Insect repellents on clothes/skin	1	never	53,378	3031.41	1.14	0.90	0.00001	0.92		53,298	48.96	0.01	0.04	0.00011	0.14	
	2	occasionally	12,289	3031.20	2.40					12,278	48.97	0.02				
	3	sometimes	5075	3033.73	3.76					5070	49.03	0.02				
	4	everyday	374	3025.88	13.31					373	49.07	0.09				
(f) Moth repellents for clothes	1	never	28,923	3031.02	1.52	0.85	<0.00001	0.92		28,876	48.94	0.01	<0.01	0.00021	<0.01	1 < 2,3
	2	sometimes	26,823	3032.20	1.57					26,793	48.98	0.01				
	3	continuously	15,370	3031.24	2.08					15,350	48.99	0.01				
(g) Fumigation insecticides	1	no	66,433	3032.27	0.99	<0.01	0.00013	0.02		66,343	48.97	0.01	0.02	0.00008	0.07	
	2	yes	4683	3020.72	3.75				2 < 1	4676	48.91	0.02				

1: Means are estimated marginal means calculated by ANCOVA; 2: SE: standard error; 3: Post hoc Bonferroni tests were applied in case of significant intergroup differences *q* < 0.05, adjusted using the Benjamini-Hochberg method for false detection rate.

**Table 3 ijerph-17-04608-t003:** Differences in neonate body sizes (weight change and length change) according to the frequency of pesticide use during pregnancy.

Pesticides	Frequency of Use			Weight Change (g)							Length Change (cm)						
				*n*	Mean ^1^	SE ^2^	*p*	*η* _p_ ^2^	*q*	Post Hoc ^3^	*n*	Mean ^1^	SE ^2^	*p*	*η* _p_ ^2^	*q*	Post Hoc ^3^
(a) Indoor insecticide sprays	1	never		46,175	1098.45	1.68	0.15	0.00008	0.32		45,755	4.28	0.01	0.01	0.00016	0.05	
	2	occasionally		18,259	1092.23	2.68					18,082	4.24	0.01				
	3	sometimes		3758	1088.45	5.89					3707	4.26	0.03				
	4	everyday		394	1095.80	17.87					388	4.30	0.08				
(b) Mosquito coils/mats	1	never		46,622	1096.67	1.68	0.79	0.00002	0.91		46,230	4.30	0.01	<0.01	0.00059	<0.01	
	2	occasionally		6531	1095.58	4.41					6460	4.23	0.02				2 < 1
	3	sometimes		9214	1097.37	3.78					9107	4.20	0.02				3 < 1
	4	everyday		6219	1091.97	4.61					6135	4.19	0.02				4 < 1
(c) Liquid insecticides	1	never		68,123	1096.20	1.35	0.41	0.00004	0.56		67,484	4.27	0.01	0.88	0.00001	0.92	
	2	occasionally		375	1105.29	18.29					363	4.30	0.08				
	3	sometimes		70	1115.26	42.26					67	4.14	0.19				
	4	everyday		18	966.49	83.22					18	4.16	0.37				
(d) Herbicides/gardening pesticides	1	never		62,543	1096.85	1.41	0.11	0.00009	0.27		61,989	4.27	0.01	0.27	0.00006	0.47	
	2	occasionally		5699	1088.23	4.75					5605	4.24	0.02				
	3	sometimes		278	1103.05	21.22					273	4.35	0.09				
	4	everyday		66	1170.25	43.46					65	4.47	0.19				
(e) Insect repellents on clothes/skin	1	never		51,440	1098.59	1.60	<0.01	0.00024	0.01		50,969	4.29	0.01	<0.01	0.00057	<0.01	
	2	occasionally		1,1904	1084.23	3.36				2 <1	11,766	4.19	0.02				2 < 1
	3	sometimes		4882	1099.50	5.28					4840	4.22	0.02				3 < 1
	4	everyday		360	1112.05	18.70					357	4.30	0.08				
(f) Moth repellents for clothes	1	never		27,737	1093.09	2.14	0.13	0.00006	0.30		27,453	4.27	0.01	0.71	0.00001	0.87	
	2	sometimes		25,961	1099.25	2.20					25,737	4.27	0.01				
	3	continuously		14,888	1096.81	2.92					14,742	4.26	0.01				
(g) Fumigation insecticides	1	no		64,080	1095.57	1.39	0.06	0.00005	0.18		63,462	4.27	0.01	0.79	<0.00001	0.91	
	2	yes		4506	1105.68	5.27					4470	4.27	0.02				

1: Means are estimated marginal means calculated by ANCOVA; 2: SE: standard error; 3: Post hoc Bonferroni tests were applied in case of significant intergroup differences *q* < 0.05, adjusted using the Benjamini-Hochberg method for false detection rate.

**Table 4 ijerph-17-04608-t004:** Associations between maternal pesticide use patterns and neonate body sizes.

Pattern No	Combination of Pesticides ^1,2^				Birth Weight (g)				Birth Length (cm)				Weight Change (g)				Length Change (cm)			
	(a)	(b)	(e)	(f)	*n*	Mean ^3^	SE ^4^	Post Hoc ^5^	*n*	Mean ^3^	SE ^4^	Post Hoc ^5^	*n*	Mean ^3^	SE ^4^	Post Hoc ^5^	*n*	Mean ^3^	SE ^4^	Post Hoc ^5^
1	−	−	−	+	15,117	3031.94	2.12		15,093	49.03	0.01		14,618	1106.43	2.96		14,519	4.34	0.01	
2	−	−	−	−	13,019	3028.64	2.28		12,999	48.94	0.02	2 < 1	12,484	1097.87	3.20		12,368	4.34	0.01	
3	+	−	−	+	4295	3027.04	3.92		4288	48.94	0.03		4162	1091.25	5.47		4124	4.27	0.02	
4	−	+	−	+	3656	3034.00	4.25		3654	48.94	0.03		3543	1093.30	5.93		3503	4.27	0.03	
5	+	+	+	+	2928	3027.78	4.81		2928	48.93	0.03		2817	1095.53	6.74		2793	4.13	0.03	5 < 1,2,3,4,6
6	+	−	−	−	2795	3036.25	4.87		2788	48.93	0.03		2679	1099.52	6.83		2657	4.38	0.03	
7	+	+	−	+	2728	3028.44	4.93		2725	48.92	0.03		2627	1088.04	6.91		2602	4.19	0.03	7 < 1,2,6
8	−	+	−	−	2532	3019.85	5.10		2531	48.93	0.03		2440	1097.03	7.15		2408	4.21	0.03	8 < 1,2,6
9	−	+	+	+	2351	3030.08	5.35		2350	48.99	0.03		2282	1095.55	7.46		2261	4.17	0.03	9 < 1,2,6
10	−	−	+	+	2307	3034.18	5.37		2304	49.02	0.03		2240	1092.17	7.49		2225	4.26	0.03	
							*p*	0.42			*p*	<0.01			*p*	0.17			*p*	<0.01
							*η* _p_ ^2^	0.00018			*η* _p_ ^2^	0.00069			*η* _p_ ^2^	0.00026			*η* _p_ ^2^	0.00181
							*q*	0.56			*q*	<0.01			*q*	0.34			*q*	<0.01

1: +: at least one used; −: never used; 2: (a) indoor insecticide sprays; (b) mosquito coils/mats; (e) insect repellents on clothes/skin; (f) moth repellents for clothes; 3: Means are estimated marginal means calculated by ANCOVA; 4: SE: standard error; 5: Post hoc Bonferroni tests were applied in case of significant intergroup differences *q* < 0.05, adjusted using the Benjamini-Hochberg method for false detection rate.

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
