# Peer review of "Association between Prenatal Exposure to Household Pesticides and Neonatal Weight and Length Growth in the Japan Environment and Children’s Study"

_ijerph, 2020, doi:10.3390/ijerph17124608_

Round 1
Reviewer 1 Report
This is a large Japanese cohort study of more than 93,000 pairs of pregnant women and their children in which various exposures have been assessed during pregnancy and their association with birth outcomes investigated. This paper focusses on prenatal exposure to household pesticides and birth weight and length and weight and length increases in the first month after birth. The authors report that “Frequencies of exposure to almost all the pesticides had no effects on birth weight and length” but “found small but significant associations between use of fumigation insecticides and decreased birth weight, and between frequencies of exposure to pyrethroid pesticides and suppression of neonatal length growth.“
These results are in a sense quite reassuring in that there are no really significant associations between pesticide use and birth weight/length. However, have the authors looked at whether pesticide use is higher in women who have low birthweight babies as a category as this is arguably a more clinically relevant outcome.
It is not immediately clear where are the results that indicate “small but significant associations between use frequencies of exposure to pyrethroid pesticides and suppression of neonatal length growth”. I presume it refers to the use of mosquito coils and so it would be better to also indicate this in the abstract. To what extent are pyrethroids also used in the other pesticide categories? For example, pyrethroids can be used in moth repellents and yet there is no association with length change and use of moth repellents for clothes.
P3 line3 97/98 “Data on the daily use of household pesticides during pregnancy were collected from questionnaires administered during the 2nd or 3rd trimesters.” This is an important point that needs further clarification as evidence suggests that trimester specific exposures may be important. How many questionnaires were from the second trimester and how many from the third? Was information from the 2nd/3rd trimesters consistent? If not, how did the authors use this data? Have the authors look specific at 2nd and 3rd trimester exposures separately?
P3 line 19: JECS profile paper appears to be reference 19 not 20 but this reference doesn’t seemingly have any material on the “classes of pesticides”.
P3 Section 2.3.3 on Covariates. Why were these specific covariates included in the analysis?
P8 Table 1 Why there is such a difference between date the questionnaire was completed and those women who never used a pesticide and those who use a pesticide at least once?
Author Response
Dear Reviewer 1
Sincerest thanks for your comments on our manuscript.
Please see the attachment.

Reviewer 2 Report
The paper is well designed, and there is a good link between the aims, goals, methods, and results/conclusions. But it is just another paper with a huge number of inclusions and a questionnaire based exposure description. We have analytical methods now to measure levels of pyrethroids in human biological materials. Why not do that in a smaller nested cohort. As it stands now we are not closer, only pointing out some associations. We can only presume the meaning of this for health outcomes.
So I want to know the hard facts here - how is the real situation? Can you create a small nested control group with actual levels of the substances of concern. I am sure that is possible. There is a trap in huge registry studies - whatever you want to find is possible with big enough N. And we want more evidence when we include the public health perspective. I wish you luck and look forward to an improved procedure!
Author Response
Dear Reviewer 2
Sincerest thanks for your comments on our manuscript.
Please see the attachment.

Round 2
Reviewer 2 Report
The paper is significantly improved. It needs a minor language edit, that can be done in-house.
Author Response
Dear Reviewer 2 Sincerest thanks for your confirmation on our manuscript. Before we resubmitted the modified version of the manuscript, we used a professional English editing service from Editage. We attached the certificate of English editing, I'd be grateful if you could confirm. Thank you again for your valuable comments and suggestions. Kind regards, Taro Matsuki
